# Speeding Up Policy Simulation in Supply Chain RL

**Vivek Farias** [1]  **Joren Gijsbrechts** [2]  **Aryan Khojandi** [3]  **Tianyi Peng** [4]  **Andrew Zheng** [5]

## Abstract

Simulating a single trajectory of a dynamical system under some state-dependent policy is a core bottleneck in policy optimization (PO) algorithms. The many inherently serial policy evaluations that must be performed in a single simulation constitute the bulk of this bottleneck. In applying PO to supply chain optimization (SCO) problems, simulating a single sample path corresponding to one month of a supply chain can take several hours. We present an iterative algorithm to accelerate policy simulation, dubbed Picard Iteration. This scheme carefully assigns policy evaluation tasks to independent processes. Within an iteration, any given process evaluates the policy only on its assigned tasks while assuming a certain 'cached' evaluation for other tasks; the cache is updated at the end of the iteration. Implemented on GPUs, this scheme admits batched evaluation of the policy across a single trajectory. We prove that the structure afforded by many SCO problems allows convergence in a small number of iterations *independent* of the horizon. We demonstrate practical speedups of 400x on large-scale SCO problems even with a single GPU, and also demonstrate practical efficacy in other RL environments.

## 1. Introduction

The core problems in supply chain optimization (SCO) all relate to managing inventory over time across some potentially large network of nodes, so as to match costly supply with uncertain demand. These problems are naturally viewed as dynamic optimization problems, albeit with intractable state-spaces that must track inventory and other resource levels across a large number of products and nodes. The development of detailed simulators ('digital twins') of supply chains in recent years has made SCO problems a ripe target for reinforcement learning. A prime example of such a target with immense economic significance is the so-called *Fulfillment Optimization* (FO) problem (Amazon.com, Inc., 2023; Acimovic & Graves, 2015; Acimovic & Farias, 2019; Zhao et al., 2022; Liu et al., 2023), which will be a main focus of this paper.

A time-step in an SCO problem typically corresponds to either a demand or supply event; there are hundreds of millions of such events in a month for a large supply chain (Shopping, 2024). As such, the task of simulating a fixed control policy is onerous, requiring the serial evaluation of the policy over a horizon, $T$, of tens or hundreds of millions of time-steps. For a policy parameterized by a non-trivial deep neural network (DNN), the task of simply simulating a *single sample path* under the policy can thus take several hours (or more) in the context of an SCO problem. This is an impediment to the application of PO methods to SCO problems which require simulating multiple sample paths of the system at each policy update iteration, and conservatively require hundreds of PO iterations to converge to a good policy. The key expensive step in any policy simulation is the task of evaluating a policy at a system state. This cannot be batched since the states encountered on a sample path are themselves computed serially. Still, one may hope that a clever scheme for parallel discrete event simulation (Fujimoto, 1990; Fujimoto et al., 2017) might help:

### 1.1. Time Warp Falls Short

Time Warp is the mainstay approach to parallel discrete event simulation (Jefferson, 1985; Jefferson et al., 1987; Ghosh et al., 1993; Barnes Jr et al., 2013; Fujimoto, 2024). Time Warp is at its core a message passing framework: a processor processes 'events' and then sends the outcome of this processing to other processors that are potentially impacted by these events. Upon receiving a processed event from a neighboring processor, a processor may need to 'roll back' its computations, if the received event invalidates them. If the set of processors potentially impacted by a given processor is small (so called 'local causality'), Time

---

*Equal contribution  [1]Sloan School of Management, Massachusetts Institute of Technology, Cambridge, MA 02139  [2]Esade Business School, Ramon Llull University, Barcelona, Spain  [3]Operations Research Center, Massachusetts Institute of Technology, Cambridge, MA 02139  [4]Columbia Business School, New York, NY 10027  [5]Sauder School of Business, The University of British Columbia, Vancouver, Canada. Correspondence to: Vivek Farias <vivekf@mit.edu>.

*Proceedings of the 42nd International Conference on Machine Learning*, Vancouver, Canada. PMLR 267, 2025. Copyright 2025 by the author(s).

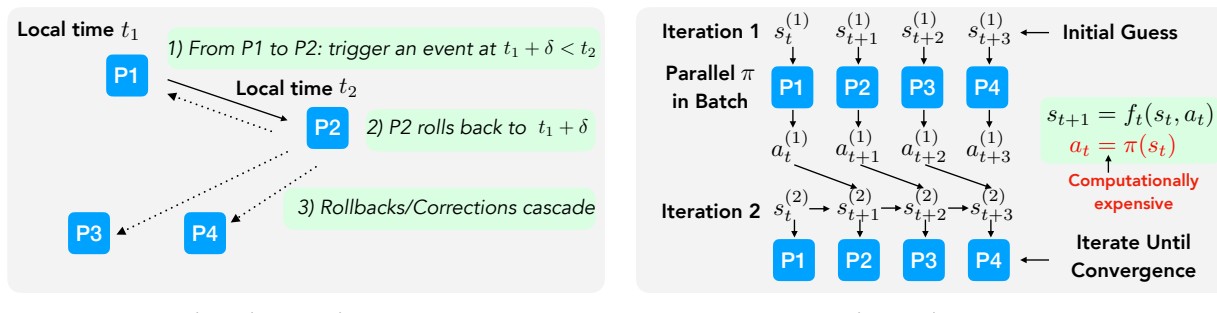

(a) Traditional Approach: Time Warp

(b) Picard Iteration

*Figure 1.* (a) Time Warp is a widely used method for simulating discrete event trajectories. It employs a message-passing algorithm where each processor (blue box) maintains a local time and processes events in parallel, potentially triggering new events. If a processor receives an event with a timestamp earlier than its local time, it must roll back, potentially causing a cascading rollback effect. This overhead makes Time Warp inefficient for general MDP trajectory simulation. (b) Instead, we observe that often for RL problems, the transition $f_t$ is computationally cheap while the policy $\pi$ is expensive. We propose Picard iteration by the following intuition: by initializing a trajectory of states (or actions), the policy $\pi$ can be executed in parallel. New trajectories are efficiently updated via lightweight transitions. This process can be iterated until convergence. Compared to Time Warp, this proposed Picard (1) is GPU-friendly, (2) enables new analyses for provable speedups, (3) achieves significant acceleration in practical SCO settings, and (4) demonstrates speedup potential for general RL.

Warp works well. But other than for special structures, this is not true for general MDPs (e.g., for SCO problems) – every processor must communicate in effect with every other processor and the resulting rollbacks reduce Time Warp to sequential evaluation while simultaneously adding message passing overhead that is highly sub-optimal for GPUs.

Indeed, it is unclear that parallelization can be fruitful for policy simulation in general, but SCO problems have two characteristics that make this a potentially feasible and worthwhile task: (1) evaluating system dynamics is substantially cheaper than policy evaluation itself, and (2) a large horizon $T$ makes the potential acceleration offered by batching significant.

So motivated, imagine, as a thought experiment, that an oracle revealed the sequence of $T$ states encountered on the sample path being simulated (i.e. what we are trying to compute in the first place). In that event policy evaluation could be trivially batched, and could even leverage the 'single program multiple data' paradigm GPUs are optimized for. Our goal here is to come close to this ideal via an iterative scheme that 'guesses' at the entire sample path being simulated, and rapidly improves this guess. See Figure 1.

### 1.2. This paper: the Picard Iteration

This paper proposes a novel iterative approach to policy simulation we dub Picard iteration[1]. Succinctly, while Picard iteration applies to general policy simulation tasks, it provably yields a speedup in the context of SCO problems; in practice we show this speedup is greater than 400x.

The Picard iteration proceeds by dividing up the simulation horizon of $T$ time-steps across (potentially, virtual) processes; the assignment of time-steps to processes may be informed by problem structure. We also initialize a 'cache' of actions, one for each time-step, that can be thought of as an initial guess of the actions that will eventually be simulated. Each process runs an independent simulation of the policy with an important tweak: a process only evaluates the policy on time-steps it has been assigned; on all other time-steps it simply uses the action for that time-step from the cache (Figure 1(b) is an example where each process is allocated a single time-step). While each process must still evaluate system transitions at every time step, these evaluations are assumed cheap relative to the cost of evaluating the policy itself. As such, each process only takes time that is roughly proportional to the number of time-steps it is assigned. All processes run in parallel. At the end of an iteration, each process updates the cache in the time-steps it was responsible for. As such, a single Picard iteration is faster than the serial policy simulation task by a factor of roughly #processes. Moreover each such iteration allows for batched policy evaluation with no message passing across processes. The key question then is how many such iterations are required to converge to the same outcome as serial policy simulation?

*Provable speedup:* We prove that in Fulfillment Optimization (FO), the number of Picard iterations required to compute the same outcome as serial policy simulation is no more than the number of nodes in the supply chain. Thus, applied to such problems the Picard iteration yields an effective speedup of $\sim$ #processes/#nodes. Since #nodes is at most a few hundred[2], and #processes can be scaled

---

[1]The proof of the Picard–Lindelöf theorem inspired our iterative framework.

[2]Amazon has $\sim$ 200 nodes (Amazon, 2024); Walmart has $\sim$

to tens of thousands via a GPU implementation, the Picard iteration guarantees a large speedup in policy simulation for those problems. In addition, Appendix B shows how our results can easily be extended to inventory control problems with replenishment.

*Practical speedup for SCO:* We show that even on a single A100 GPU, the use of the Picard iteration yields an effective speedup of greater than $400x$ relative to serial policy evaluation for a large-scale FO problem. The speedup relative to a (tailored, highly optimized) Time Warp algorithm is approximately 100x. The speedup displays attractive linear scaling with the number of processes (as predicted by our theory) and is *robust* across a range of challenging problem instances. In addition, this speedup is persistent when scaling to an end-to-end RL pipeline.

*Practical speedup for RL problems beyond supply chain:* In general MDP problems, we show Picard converges to the sequential simulation output after no more than $T$ iterations. Given its correctness in general, we explore whether the number of Picard iterations required is a lot smaller than $T$ for problems outside SCO. Here we show that for a majority of OpenAI Gym MuJoCo environments, the Picard iteration could potentially yield a speedup of up to 40x.

### 1.3. Related literature

*SCO and RL:* Supply Chain Optimization (SCO) problems represent a classic family of dynamic optimization problems with intractable state-spaces, and are becoming increasingly important due to the growing volumes in e-commerce. For instance, a 1% cost reduction in fulfillment for Amazon can be translated to about 1B US dollars savings (Amazon.com, Inc., 2023). Applying Deep Reinforcement Learning (RL) to solve intractable SCO problems has garnered increasing interest in recent years (Oroojlooyjadid et al., 2022; Gijsbrechts et al., 2022; Temizöz et al., 2025; Alvo et al., 2023). Large companies such as Amazon (Madeka et al., 2022), Alibaba (Liu et al., 2023), and JD.com (Qi et al., 2023) have reportedly been testing RL at scale in SCO contexts. Our motivation for this work is to eliminate the high costs of training and back-testing in these problems due to long horizons, $T$. While our framework is general, we showcase it on a representative problem in SCO: online fulfillment optimization (Acimovic & Farias, 2019; Zhao et al., 2022; Xu et al., 2020; Talluri & Van Ryzin, 1998; Ma, 2023; Amil et al., 2022; Acimovic & Graves, 2015; Jasin & Sinha, 2015; Andrews et al., 2019).

*Parallel RL:* There is a substantial body of literature on parallel reinforcement learning (see e.g., Asynchronous RL (Mnih et al., 2016), RLlib (Liang et al., 2018), Envpool (Weng et al., 2022), and others (Stooke & Abbeel, 2018;

30 nodes in the U.S (Walmart, 2022).

Clemente et al., 2017; Petrenko et al., 2020; Rutherford et al., 2023)), where embarrassing parallelism is implemented across different agents or rollouts. In contrast Picard iteration is concerned with an orthogonal problem: accelerating the evaluation of a single trajectory, which is particularly relevant for our SCO settings where $T$ is gigantic. For such problems Picard iteration *complements* parallel RL.

*Parallel Discrete Event Simulation:* As discussed earlier, Time Warp (Jefferson et al., 1987; Steinman, 1993; Barnes Jr et al., 2013; Fujimoto et al., 2017; Fujimoto, 2024), a mainstay parallel discrete event simulation algorithm could reasonably be considered an alternative to our proposal; in fact Time Warp has recently been optimized for GPUs (Liu & Andelfinger, 2017). As discussed, SCO problems (and we anticipate many RL problems in general) do not satisfy the local causality assumptions needed for Time Warp to work well (Radhakrishnan et al., 1996). We implement an optimized variant of Time Warp suitable to GPUs and show that Picard iteration is almost two orders of magnitude faster that Time Warp. We attribute this to the inability to re-use policy evaluations computed on incorrect states within Time Warp, a point that will become clear later.

Picard iteration leverages the batch computation power of GPUs for sequential problems. Non-RL examples in this spirit include using lookahead decoding for LLM inference (Fu et al., 2024) and a three-stage algorithm for simulating cellular base stations (Li et al., 2013). A distinctive feature of our approach is the theoretical guarantee of speedup, where the number of required Picard iterations captures how coupled a problem is. *This is rarely seen in the parallel computing literature and may be of broad interest.*

## 2. Model and algorithm

We consider a dynamical system with general state-space $\mathcal{S}$, action space $\mathcal{A}$, and a disturbance space $\Omega$. The dynamical system itself is specified by a function $f : \mathcal{S} \times \mathcal{A} \times \Omega \to \mathcal{S}$. We assume the existence of an 'always feasible' action $a^\phi$, so that for all $s$, $a^\phi \in \mathcal{A}(s)$, the set of feasible actions at state $s$. For our purposes a policy is simply a map $\pi : \mathcal{S} \times \Omega \to \mathcal{A}$; one may think of $\pi(\cdot)$ as a DNN and assume $\pi(s) \in \mathcal{A}(s) \; \forall s$. We may think of the disturbance here as capturing both exogenous shocks (e.g., demand in an SCO problem), as well as any randomization endogenous to $\pi$.

We next formally define the task of policy simulation. As input, we are given an initial state $s_1 \in \mathcal{S}$, a horizon $T$, and a sequence of disturbances $\{\omega_t : t \in [T]\}$. Our desired output is the sample path of actions under $\pi$, $\{a_t^{\text{seq}} : t \in [T]\}$ defined according to $a_t^{\text{seq}} = \pi(s_t, \omega_t)$, where $s_{t+1} = f(s_t, a_t^{\text{seq}}, \omega_t)$. We remark that we view the application of $\pi(\cdot)$, i.e. the computation $\pi(s, \omega)$ as computationally costly, while we view the application of $f(\cdot)$

given an action, $f(s, a, \omega)$ as computationally cheap. This is certainly the case in RL for SCO problems. Assuming a single application of $\pi(\cdot)$ takes unit time, our desire is to compute $\{a_t^{\text{seq}}\}$ in time $\ll T$.

We next present the Picard iteration (Algorithm 1). We assume $M$ virtual processes indexed by $m$. Each $m$ is assigned a disjoint partition of time-steps, $\mathcal{T}_m \subset [T]$ where $\cup_m \mathcal{T}_m = [T]$. Algorithm 1 implicitly assumes the 'cache' of actions, $\{\alpha_t^k\}$ and the disturbance sequence $\{\omega_t\}$ sit in shared memory. Several remarks are in order:

**Correctness:** It is easy to see that the Picard iteration outputs the correct sequence of actions, i.e. $\{a_t^{\text{seq}}\}$ in at most $T$ iterations. Observe that for the processor responsible for time-step $t = 1$, say $m$, $a_1^{1,m} = a_1^{\text{seq}}$. Consequently, $\alpha_1^1 = a_1^{\text{seq}}$. Thus, at iteration $k = 2$, the processor responsible for $t = 2$, say $\tilde{m}$, will take the correct sequential action at $t = 1$ and thus the correct sequential action at $t = 2$, i.e. $a_2^{2,\tilde{m}} = a_2^{\text{seq}}$, so that $\alpha_2^2 = a_2^{\text{seq}}$. Continuing in this fashion, we can show:

**Proposition 2.1.** *The Picard iteration converges in at most $T$ iterations and returns $\{a_t^{\text{seq}}\}$.*

While this proposition shows the correctness of the algorithm, it is not useful: if we required $T$ iterations for convergence, we would achieve no speedup. We will later prove that Algorithm 1 converges in a small number of iterations *independent of $T$* in a large class of SCO problems.

**Speedup and batching:** A single iteration of Algorithm 1 achieves substantial speedup over sequential computing. Specifically, assume that time-steps are divided up equally across all processes so that $|\mathcal{T}_m| \sim T/M$ for all $m$. Then, under the assumption that the time to evaluate $\pi(\cdot)$ is much larger than the time to evaluate $f(\cdot)$, this speedup is approximately $M$. Consequently, the effective speedup provided by Algorithm 1 is $M/\#\text{iterations}$. It is also worth noting that Algorithm 1 allows for the batched application of $\pi(\cdot)$. Specifically, the first application of $\pi(\cdot)$ on each of the $M$ processes can be batched together, following which the second application of $\pi(\cdot)$ on each of these processes can be batched, and so forth.

Our discussion so far applies to general dynamical systems. We cannot hope for an effective speedup in this generality. The next Section will focus on a large class of SCO problems, where we will theoretically establish that Algorithm 1 achieves a non-trivial speedup over sequential computation.

## 3. Picard Iteration and RL for Fulfillment Optimization

We focus here on the *Fulfillment Optimization* (FO) problem, a central class of SCO problems that has recently attracted significant attention for RL solutions (Amazon.com, Inc.,

2023; Acimovic & Farias, 2019; Amil et al., 2022; Acimovic & Graves, 2015; Andrews et al., 2019). We also applied Picard to inventory control with replenishment, another representative class of SCO problems (Alvo et al., 2023; Gijsbrechts et al., 2022; Madeka et al., 2022; Dehaybe et al., 2024; Liu et al., 2023), demonstrating favorable theoretical and practical speedups. Details are deferred to Appendix B.

For FO, We are concerned with $I$ *products*, indexed by $i \in [I]$. Inventory of each product is carried at one or more of $J$ *nodes*, indexed by $j \in [J]$. Each node is endowed with a processing capacity. Let $c_{t,j} \in \mathbb{R}$ be the remaining capacity of node $j$ at time $t$. In addition to the capacity constraint, let $x_{t,i,j} \in \mathbb{R}$ be the inventory level of product $i$ at node $j$ and time $t$. We use $c_t \in \mathbb{R}^J$ and $x_t \in \mathbb{R}^{IJ}$ to simplify the notation. The state at time $t$ is then $s_t := (x_t, c_t)$. An *order* at time $t$ is associated with a product $i(\omega_t)$ and a reward vector $r(\omega_t) \in \mathbb{R}^J$. A policy $\pi$ assigns this order to a node with available inventory of product $i(\omega_t)$ and non-zero capacity and earns the corresponding reward (or it does not fulfill and earns zero reward):

$$a_t := \pi(s_t, \omega_t) \in \{j \in [J] \mid c_{t,j} > 0, x_{t,i(\omega_t),j} > 0\} \cup \{0\}$$

where '0' is the action of not fulfilling, i.e., $a^\phi$. The goal is to design a policy that maximizes the total reward over some finite horizon $T$.

The FO problem has state space $\mathcal{S} = \mathbb{R}^{IJ} \times \mathbb{R}^J$ and action space $\mathcal{A} = [J] \cup \{0\}$. Capacity is updated according to $c_{t+1} = c_t - e_{a_t}$ while available inventory is updated according to $x_{t+1} = x_t - e_{i(\omega_t),a_t}$ (where $e_j$ denotes the $j$th unit vector); this specifies $f(\cdot)$, which is computationally trivial.

There is just one design decision we need to make in considering how to apply the Picard iteration to the FO problem, viz. how to assign time-steps to each of the $M$ processes. We consider the approach of partitioning by products: all time-steps associated with a given product are assigned to the same process. Specifically, let $T_i = \{t : i(\omega_t) = i\}$. Let $[I]$ be divided into $M$ disjoint partitions; denote the $m$th $I_m$. Let processor $m$ be responsible for all time-steps associated with products in $I_m$, i.e., , $\mathcal{T}_m = \cup_{i \in I_m} T_i$. Ideally, we find a partition of $[I]$ such that each $\mathcal{T}_m$ is roughly the same size. With this setup, we now state the main result of this section informally:

**Theorem 3.1** (Informal). *Provided $\pi(\cdot)$ satisfies a set of regularity conditions, the Picard iteration, Algorithm 1, converges in at most $J + 1$ iterations for the FO problem.*

**Speedup:** A corollary of the above result is that Algorithm 1 provides an effective speedup of $T/(J \max_m |\mathcal{T}_m|)$. When tasks are constant sizes, $T/\max_m |\mathcal{T}_m| \sim M$ (see e.g., 'balls into bins' problems (Raab & Steger, 1998)) so that the speedup provided by Algorithm 1 is $\sim M/J$. As noted in the introduction, even for a large instance of the FO problem

---

**Algorithm 1** The Picard Iteration

---
0: $k \leftarrow 1$, $\alpha_t^0 \leftarrow a^\phi$ for $t \in [T]$ {Initialize cache to always feasible action $a^\phi$}
0: **while** true **do**
0:     **parfor** $m \leftarrow 1$ to $M${Evaluate each process in parallel}
0:         $s_1^{k,m} \leftarrow s_1$
0:         **for** $t \leftarrow 1$ to $T$ **do**
0:             **if** $t \in \mathcal{T}_m$ **then** $a_t^{k,m} = \pi(s_t^{k,m}, \omega_t)$ {Process $m$ evaluates $\pi(\cdot)$}
0:             **else** $a_t^{k,m} = \alpha_t^{k-1}$ if $\alpha_t^{k-1} \in \mathcal{A}(s_t^{k,m})$ (or $a^\phi$ o.w.) {Process $m$ uses cached action}
0:             $s_{t+1}^{k,m} = f(s_t^{k,m}, a_t^{k,m}, \omega_t)$ {Process $m$ updates its state}
0:         $\alpha_t^k = a_t^{k,m}$ **for** $m \leftarrow 1$ to $M$, $t \in \mathcal{T}_m$ {Process $m$ updates cache on $t \in \mathcal{T}_m$}
0:     **if** $\alpha_t^k = \alpha_t^{k-1}$ $\forall t$ **then return** $\{\alpha_t^k\}$ {Converged}
0:     **else** $k \leftarrow k + 1$
     =0

---

(corresponding say to a retailer like Amazon), $J \sim 200$, whereas we can take $M \sim 10^4$ to $10^5$ so that we can hope for a speedup on the order of $10^2$ to $10^3$.

### 3.1. Proof of special case

We present here a short proof of a special case of Theorem 3.1, which should serve to provide some intuition about the result and its proof. Theorem 3.1 will also hold for a much more general class of policies, which we make precise in Theorem 3.2; however we defer this proof to the Appendix. For the special case we consider, we assume that (1) inventory is not a constraint (or equivalently that $x_1 = T\mathbf{1}$), and (2) that the policy $\pi(\cdot)$ is greedy (greedy policies have been used widely in practice (Zhao et al., 2022; Xu et al., 2020)) so that $\pi(s, \omega) \in \arg\max_{j:c_j > 0} r_j(\omega)$.

The crux of the proof lies in understanding the structural properties of 'wrong' actions. To show that $a_t^k$ is correct (i.e., $a_t^k = a_t^{\text{seq}}$), we need to show that the mistakes made in round $k-1$ by other processors are constrained in a desired way (they would impact $s_t$ and, therefore, $a_t^k$). We conduct an inductive proof for showing those properties, where the challenges lie in identifying the 'right' induction hypothesis and connecting different structural properties.

For the special greedy case, the 'right' structure properties turn out to be related to the set of nodes that run out of capacity in the sequential scenario. In particular, denote by $\tau_j$ the first time at which node $j$ runs out of inventory assuming sequentially correct actions: $\tau_j = \min\{t : c_{t,j}^{\text{seq}} = 0\}$ (or $T + 1$ if $c_{t,j}^{\text{seq}} > 0$ for all $t \in [T]$). Next we define $\mathcal{Q}_t$ to be the set of all nodes that have run out of capacity at some time $t' \leq t$: $\mathcal{Q}_t = \{j \in [J] : \tau_j \leq t\}$. We then have the following invariant:

**Lemma 3.2.** *For any $t \in [T]$ and all iterations $k$ of Algorithm 1, we have that the cached action $\alpha_t^k \in \{a_t^{\text{seq}}\} \cup \mathcal{Q}_t$.*

That is to say, the action taken at time $t$ is either going to be correct or will go to a node that would run out of

capacity in the sequential scenario. Assuming Lemma 3.2, we prove Theorem 3.1 for the special case. Let us denote by $\tilde{\tau}_1, \tilde{\tau}_2, \ldots, \tilde{\tau}_J$, the values of $\tau_j$ sorted from smallest to greatest. We establish by induction that in iteration $k$, $\alpha_t^k = a_t^{\text{seq}}$ for all $t < \tilde{\tau}_k$, i.e. the cached action is correct for all times $t < \tilde{\tau}_k$. This establishes a stronger statement: we converge after $|\mathcal{Q}_T| + 1$ iterations.

The induction statement is vacuously true for the initial cache; assume the statement true up to some $k$, and consider $k+1$. By the induction hypothesis, at iteration $k+1$, all processes $m$ have access to the correct cached action for times $t < \tilde{\tau}_k$ so that $a_t^{k+1,m} = a_t^{\text{seq}}$ for $t < \tilde{\tau}_k$. Consequently, $c_t^{k+1,m} = c_t^{\text{seq}}$ for $t \leq \tilde{\tau}_k$, and in particular, $c_{\tilde{\tau}_k,j}^{k+1,m} = 0$ for all $j \in \mathcal{Q}_{\tilde{\tau}_k}$. Now by Lemma 3.2, we must have that for any $t \in \mathcal{T}_m$ such that $\tilde{\tau}_k \leq t < \tilde{\tau}_{k+1}$,

$$a_t^{k+1,m} \in \{a_t^{\text{seq}}\} \cup \mathcal{Q}_t = \{a_t^{\text{seq}}\} \cup \mathcal{Q}_{\tilde{\tau}_k}.$$

But since $c_{\tilde{\tau}_k,j}^{k+1,m} = 0$ for all $j \in \mathcal{Q}_{\tilde{\tau}_k}$, it must be that $a_t^{k+1,m} = a_t^{\text{seq}}$, so that $\alpha_t^{k+1} = a_t^{\text{seq}}$ for $\tilde{\tau}_k \leq t < \tilde{\tau}_{k+1}$ proving the inductive step and completing the proof. We finish up with proving Lemma 3.2.

**Proof of Lemma 3.2:** Assume the Lemma holds for all $t' \in [T]$ for iterations up to $k - 1$, and for $t' \leq t$ in iteration $k$. Consider time-step $t + 1$ then and assume that this is handled by processor $m$. If $a_{t+1}^{k,m} \in \mathcal{Q}_{t+1}$ we are done, and so assume that $a_{t+1}^{k,m} \in [J] \backslash \mathcal{Q}_{t+1}$. Now $a_{t+1}^{\text{seq}} \in \arg\max_{[J] \backslash \mathcal{Q}_{t+1}} r_j(\omega_{t+1})$. So if $c_{t+1,j}^{k,m} > 0$ for all $j \in [J] \backslash \mathcal{Q}_{t+1}$, we are done. Now assume that $a_{t'}^{k,m} = j \in [J] \backslash \mathcal{Q}_{t+1}$ for some $t' \leq t$. By the induction hypothesis, it must be that $j \in \{a_t^{\text{seq}}\} \cup \mathcal{Q}_{t'}$. But $\mathcal{Q}_{t'} \subseteq \mathcal{Q}_{t+1}$, and since $j \in [J] \backslash \mathcal{Q}_{t+1}$ by assumption, it must then be that $j = a_t^{\text{seq}}$. We have consequently shown:

$$c_{t+1,j}^{k,m} = c_{1,j} - \sum_{t' \leq t} \mathbf{1}\{a_{t'}^{k,m} = j\}$$

$$\geq c_{1,j} - \sum_{t' \leq t} \mathbf{1}\{a_{t'}^{\text{seq}} = j\} = c_{t+1,j}^{\text{seq}} > 0$$

for $j \in [J] \backslash \mathcal{Q}_{t+1}$ completing the proof.

## 3.2. The general setting

We now remove the restrictions considered in the special case above, allowing for arbitrary initial inventories (so that inventory feasibility matters) and consider a more general class of state-dependent policies. Specifically, we require that the policy $\pi(\cdot)$ satisfy the following assumptions:

**Assumption 1 (Inventory Independence):** Fix some $i \in [I]$ and let $x$ and $x'$ be two inventory positions in $\mathbb{R}^{IJ}$ such that $x_{i,j} = x'_{i,j}$ for all $j$. Then, if $i(\omega) = i$, we must have $\pi((x,c),\omega) = \pi((x',c),\omega)$. In simple terms, the fulfillment decision for an order of a specific product depends only on the inventory position across all nodes of that product.

**Assumption 2 (Consistency):** $\pi((x,c),\omega) = j$, and let $i(\omega) = i$. Consider a distinct state $(x',c')$ such that $x'$ differs from $x$ only in its $(i,j')$th component, and $c'$ differs from $c$ only in its $j'$ component. Then, we must have $\pi((x',c'),\omega) \in \{j,j'\}$.

**Assumption 3 (Monotonicity):** Let $\pi((x,c),\omega) = j$, and let $i(\omega) = i$. Then, if $x' = x + e_{i,j}$, $\pi((x',c),\omega) = j$, i.e. increasing inventory of product $i$ to the node it was originally fulfilled from by $\pi$ will not change the fulfillment decision. Similarly, let $c' = c + e_{j'}$ for any $j'$ such that $c_{j'} > 0$. Then, $\pi((x,c'),\omega) = j$. Specifically, adding capacity to any node with positive capacity will not change the fulfillment decision.

The Assumptions 1-2 are natural and met by most policies proposed for the FO problem thus far. The Assumption 3 is also met by perhaps the most important class of policies proposed for the FO problem, the so-called 'bid price' policies (Talluri & Van Ryzin, 1998; Andrews et al., 2019)[3]. Having stated these requirements, we can now re-state a refined version of Theorem 3.1 formally.

**Theorem 3.2.** *Provided $\pi(\cdot)$ satisfies Inventory Independence, Consistency and Monotonicity, Algorithm 1 converges in at most $|\mathcal{Q}_T| + 1$ iterations for the FO problem.*

**The proof of Theorem 3.2** is more advanced due to the broader class of policies we allow. As the policy can depend on the inventory in a general way, Lemma 3.2 no longer holds. Instead, we find that the inventory levels maintain certain monotonicity. The formal proof is provided in the Appendix A.2. Recall that $|\mathcal{Q}_T|$ represents the number of nodes that reach their capacity limit under the assumption of sequentially correct actions. In scenarios where demand is less than supply, $|\mathcal{Q}_T|$ can be significantly less than $J$. Consequently, Theorem 3.2 can be interpreted as an

---

[3]We experimentally demonstrate the robustness of Picard when Assumption 3 violated; see Appendix.

*instance-dependent* bound for Picard iterations.

In addition, we also generalize the theorem to the scenarios where replenishment is allowed (see Appendix C).

## 4. Experiments with Fulfillment Optimization

This section presents the results of an implementation of the Picard iteration on large scale FO problem instances. *We also conducted experiments on Inventory Control problems, another representative class of SCO problems. The results are qualitatively similar (see Appendix B).* The source code has been made publicly available (see the supplementary for anonymity). We seek to establish a few key points:

- **Policy Evaluation Speedup.** The Picard iteration offers a speedup of **350-450x** over sequential simulation in the FO problem. The speedup scales approximately linearly with batch size and is robust to a variety of problem characteristics.

- **Policy Optimization Speedup.** An end-to-end implementation of a policy gradient algorithm on the FO problem highlights the value of our speedup: on a large scale instance, time to convergence with sequential evaluation is $\sim$**10h**; speeding up policy evaluation on that instance with Picard iteration yields the same policy in **2mins**.

- **Message Passing Degrades Performance.** A well regarded scheme for parallel discrete event simulation is **Time Warp**. Carefully adapting Time Warp to the FO problem yields only a 5x improvement over sequential simulation assuming special structure; on those problems, Picard is **88x** faster than Time Warp.

**Implementation.** We implement two practical optimizations that improve performance (but do not impact our theoretical analysis): First, if $t_{\text{reset}}$ is the smallest $t$ for which $\alpha_t^k \neq \alpha_t^{k-1}$, then we know that the actions evaluated for times $t < t_{\text{reset}}$ are correct and it is sufficient to start the $k$th Picard iteration at time $t = t_{\text{reset}}$. Second, as opposed to running Picard iteration over the entire horizon, we run the iteration in 'chunks' of size `max_steps` and move on to the next chunk only after convergence of the preceding one. More precisely, we run the **for** loop in Line 5 of the algorithm over $t \in [t_{\text{reset}}, \min(T, t_{\text{reset}} + \texttt{max\_steps})]$. Tuning the `max_steps` parameter thus trades off the need for synchronization (the number of iterations of the **while** loop in Line 2), with the potential for 'wasting' computation (the number of iterations of the **for** loop in Line 5). Tuning `max_steps` is straightforward: a reasonable heuristic is to simulate a sub-trajectory of length $\ll T$, and select the values which minimize the time required to simulate this sub-trajectory. All experiments were conducted on a single

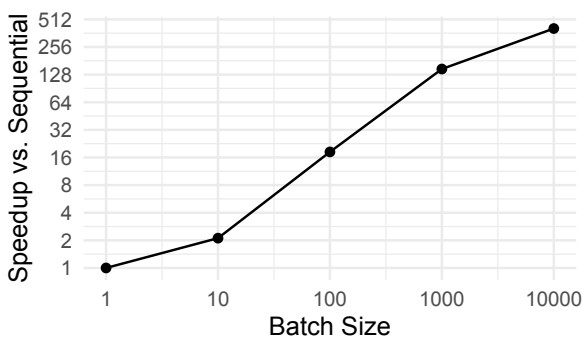

*Figure 2.* Picard runtime on problem instances with uniformly distributed demand, as a function of the number of processes (batch size $M$). The $y$-axis normalizes computation time to $M = 1$ (i.e., speedup). For $M = 1e4$, we achieve a $441\times$ speedup relative to the sequential algorithm.

A100 GPU with 40GB of VRAM. The code is implemented in JAX (Bradbury et al., 2018) and is available on Github[4]. See further details in Appendix A.3.

### 4.1. Policy evaluation speedup

For policy evaluation, we approximate a greedy-like policy using a simple MLP with two layers of width 64, with the goal of constructing a policy with predictable behavior which represents a realistic computational workload. We evaluate problem instances with $J = 30$ nodes, $I = 1M$ products, and $T = 3M$ orders, which represent a (moderately) large-scale real-world problem (Amazon, 2025; Walmart, 2022). We conduct two sets of experiments:

**Uniform Demand, Increasing Batch Size:** We divide the $T$ orders across the $I$ products uniformly at random, and vary the batch size $M$. We determined that our choice of $\pi(\cdot)$ allowed for a batch size of up to 1e4 and thus vary $M$ in factors of 10 up to 1e4. Products are assigned to each partition $\mathcal{T}_m$ at random. Recall from our discussion in Section 3, we expect a speedup of at least $O(M/J)$.

Figure 2 plots our speedup relative to the sequential baseline. We see in all cases a speedup greater than $M/J$; in the case of $M = 1e4$ this translates to an actual speedup of **441x**. We observe further that the speedup is largely linear with respect to $M$, but flags as $M$ grows large. We attribute this to an increased number of iterations required for convergence as the number of products assigned to a single partition $\mathcal{T}_m$ decreases.

**Heavy-Tailed Demand, Maximum Batch Size:** We next consider that the $T$ orders are *not* split uniformly across the $I$ products, but rather that a large fraction of the orders

---

[4] https://github.com/atzheng/picard-iteration-icml

| $\beta$ | Product Partitions | Uniform Partitions |
|---|---|---|
| 0.0 | **441** | 363 |
| -0.2 | **431** | 360 |
| -0.4 | 194 | **361** |
| -0.6 | 154 | **358** |
| -0.8 | 24 | **354** |
| -1.0 | 5 | **355** |

*Table 1.* Speedup of Picard Iteration relative to sequential, as a function of the demand distribution.

are covered by a disproportionately small fraction of the products. We model this as $Q_i \propto i^\beta$, with typical values of $\beta$ between -0.6 and -0.8 (Brynjolfsson et al., 2003; 2010). The case $\beta = 0$ corresponds to uniform demand.

We expect this setting to be more challenging. If all orders for a product were assigned to the same partition $\mathcal{T}_m$, partitions would be highly imbalanced, bottlenecking line 3 of the algorithm and limiting batching opportunities. A simpler alternative is to assign each of the $T$ orders randomly to a partition, ensuring uniform sizes but likely increasing conflicts (e.g., multiple partitions acting on the same inventory). We evaluate both approaches in Table 1, finding that uniform assignment dominates for larger $\beta$ (heavy-tailed demand). Regardless of $\beta$, we achieve at least **350x** speedup.

### 4.2. End-to-end policy optimization for FO

Next, we turn to finding an optimal policy for a sequence of FO instances using a policy gradient approach. At each iteration the approach rolls-out the policy at that iterate and computes a policy gradient. We will consider the speedup achieved by computing this roll-out via the Picard iteration, as compared with simply rolling it out sequentially.

In greater detail, we consider the following policy parametrization: let $f_\theta(s) \triangleq [\mu_{inv}^\theta(s), \mu_{cap}^\theta(s)]$ with $\mu_{inv}^\theta(s) \in \mathbb{R}^{IJ}, \mu_{cap}^\theta(s) \in \mathbb{R}^J$, be some function of state parameterized by $\theta$; here we take $f_\theta$ to be a two layer MLP of width 64. We then consider the dual policy (Balseiro et al., 2023)

$$\pi_\theta(s_t, \omega_t) \in \arg\max_j (r_j(\omega_t) - \mu_{inv,(i(\omega_t),j)}^\theta(s_t) - \mu_{cap,j}^\theta(s_t)).$$

For optimizing $\theta$, we perform 1K gradient steps using Adam with learning rate 3e-3. Table 2 reports our results for a sequence of problems of increasing size. In each experiment, we report overall runtime for both sequential policy roll-out and roll-out via the Picard iteration. For the problem instance with 3M orders, the sequential implementation takes approximate **10hrs** of wall clock time; whereas using the Picard iteration requires less than **2mins**, demonstrating the value of Picard iteration for speeding up RL in SCO problems.

| Problem Scale #Orders/ #Products | Runtime Sequential | Runtime Picard | Picard Speedup |
|---|---|---|---|
| $30k/10k$ | 6m55s | 1m01s | 6.8x |
| $300k/100k$ | $\sim 1$h | 1m04s | 56.25x |
| $3M/1M$ | $\sim 10$h | 1m39s | 363x |

*Table 2.* Runtimes for an end-to-end RL pipeline.

### 4.3. A comparison to Time Warp on the GPU

Surprisingly, there is no obvious baseline approach for the parallel policy simulation task. As detailed in Section 1.1, one popular family of approaches to parallel discrete event simulation is the Time Warp algorithm (Fujimoto et al., 2017; Jefferson et al., 1987; Steinman, 1993; Barnes Jr et al., 2013) which recently has been adapted to GPUs (Liu & Andelfinger, 2017).

In attempting to generalize Time Warp, at least to the FO problem, we make the following observation: suppose a policy for FO satisfies the regularity conditions outlined in Section 3.2 (Assumptions 1, 2 and 3), and suppose all processes have the correct state at time $t_0$. Then, we observe that all processes can execute in parallel between $t_0$ and $t_0 + \min_j c_{t_0, j}$ without requiring a rollback. We exploit this to implement Time Warp efficiently on GPU, in analogy to (Liu & Andelfinger, 2017).

We benchmark this approach in the setting with $3M$ orders and $1M$ products. Recall that Picard Iteration achieves its 441x speedup over sequential simulation in this section (Section 4.1). In contrast we observe that Time Warp achieves a speedup of **5x**. Picard is thus **88x** faster than Time Warp, even when the latter is specifically tuned to exploit problem structure.

## 5. Exploratory Experiments on RL problems beyond Supply Chain

While we have focused our experiments and analysis on SCO problems, **our Picard iteration applies to general RL environments.** This Section briefly explores applying the Picard iteration to RL environments outside SCO. Specifically, we consider a variety of OpenAI gym MuJoCo environments commonly used for RL benchmarks, implemented in `jax` via the Brax library (Freeman et al., 2021). We consider using the Picard iteration for policy roll-out at each policy optimization iteration in place of sequential simulation. We use PPO (Schulman et al., 2017), although our simulation approach is agnostic to the learning algorithm. Our goal here is simple: we wish to show that Picard iteration converges in a small number of iterations, $\ll T$.

We adopt the network architecture for $\pi$, and most policy optimization hyperparameters from the CleanRL benchmark

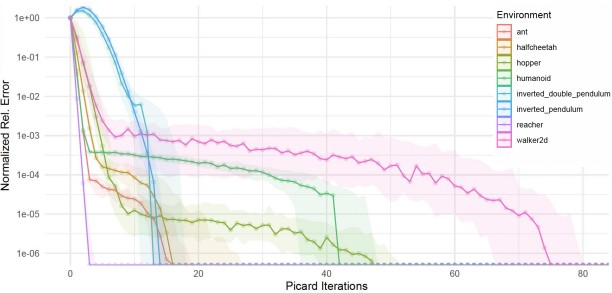

*Figure 3.* Convergence of the Picard iteration for Gym MuJoCo environments, measured in relative RMSE between the Picard trajectory and the sequentially simulated correct trajectory (normalized by RMSE of the draft trajectory $\{s_t^0\}_{t \in [T]}$). Solid line shows median RMSE at each iteration over 30 seeds; error bars show 20th and 80th percentiles. Median rel. RMSE converges to $\leq 0.1\%$ in under fifteen iterations for all environments, whereas $T = 200$; five of eight converge within 5 iterations.

(Huang et al., 2022), a popular baseline implementation for RL algorithms. Jax training code is adapted from (Lu et al., 2022), which tries to replicate CleanRL functionality. For numerical stability, we set a learning rate of 3e-5 and perform one update per trajectory instead of 10.

**Setup:** Our goal is to measure the speedup afforded by Picard iteration to the policy evaluation step in a given PPO iteration. For each environment, we run PPO for a total of 100k time-steps with policy updates occurring every 2048 time-steps. We collect the final and penultimate policy iterates which we refer to as $\pi^+$ and $\pi^0$ respectively. Our goal is to simulate $\pi^+$ via Algorithm 1. We choose to set $M = T = 200$, such that each $\mathcal{T}_m$ consists of a single time-step. There is a natural and attractive choice of the initial cache: specifically, we set $\alpha_t^0 = \pi^0(s_t^0, \omega_t)$, the sequence of actions taken by the previous policy iterate on the sample path in question.

**Results:** Figure 3 plots the convergence of the Picard iteration across several environments, and multiple seeds for each environment. Defining the state trajectory simulated at the $k$th Picard iteration according to $s_{t-1}^k = f(s_t^k, \alpha_t^k, \omega_t)$, we measure convergence here via relative root-mean-squared-error (RMSE): $\sum_t \|s_t^{+,\text{seq}} - s_t^k\|_2 / \sum_t \|s_t^{+,\text{seq}}\|_2$. where $s_t^{+,\text{seq}}$ is the 'correct' state under $\pi^+$ at $t$. This quantity is guaranteed to be 0 (up to numerical precision) for $k = T = 200$ by Proposition 2.1, but the key question is whether we get convergence in $k \ll T$.

Figure 3 answers this question in the affirmative. We see that in five of eight environments, the relative RMSE (median over 30 random seeds) converges to $\leq 0.1\%$ in **under five iterations** whereas $T = 200$; further, all environments converge within fifteen iterations. These are exciting results suggesting the Picard framework might be of value in gen-

eral environments as well. For instance if evaluation of $f$ were sufficiently faster than evaluation of $\pi$ (i.e. if dynamics were cheaper to simulate than policy evaluation), the results here would lead to an end-to-end speedup of **13-40x**.

## 6. Future Work

There are exciting avenues for future work on the surprisingly under-explored problem of parallel policy simulation:
**SCO Problems:** Our findings extend to other SCO problems. In Appendix B, we show that Picard iteration achieves comparable empirical and theoretical speedups when applied to the *One Warehouse Multi-Retailer* replenishment problem. Many more opportunities for exploration remain.
**Convergence in Other Environments:** We showed promising results for environments outside SCO (viz. MuJoCo), although the theoretical basis for Picard's success in these settings remains unclear. Analyzing the factors influencing convergence in different environments would be a valuable next step, perhaps starting with simple time-varying linear systems. In 'contractive' systems, Picard iteration should converge linearly, with the rate determined by the associated contraction factor.
**Overall Speedup in Other RL problems:** Assuming a batch size $M$ and that we require $K$ Picard iterations for convergence, the overall speedup is $(\eta + 1)/(\eta + 1/M)K$. Here $\eta$ is the ratio of the amount of time it takes to evaluate $f(\cdot)$ to that of evaluating $\pi(\cdot)$. In the case of SCO problems this ratio is $\sim 0$. On the other hand, for environments like MuJoCo, this ratio is actually close to $1/5$. As such, we see that speeding up $f$ can result in dramatic time savings in concert with Picard. In the case of MuJoCo, this could be achieved with a more efficient vectorized physics engine.

## Impact Statement

This paper presents work whose goal is to advance the field of Machine Learning. There are many potential societal consequences of our work, none which we feel must be specifically highlighted here.

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

# A. Appendix / supplemental material

## A.1. Proof of Proposition 2.1

We will prove the proposition by induction on the following hypothesis:

$$\alpha_t^k = a_t^{\text{seq}} \quad \forall t \le k,$$

whence the desired result will follow immediately by setting $k = T$. Letting $\tilde{m}_t^k$ denote the processor to which order $t$ is assigned in iteration $k$, we have $\alpha_1^1 = a_1^{1,\tilde{m}_1^1} = a_t^{\text{seq}}$, since all processors have the correct state at the beginning of the horizon: $s_1^{1,m} = s_1 \quad \forall m$.

Now, assume that $\alpha_t^k = a_t^{\text{seq}} \quad \forall t \le k$. By this induction hypothesis (and the state update defined in Section 3), we have $s_k^{k,m} = s_k \quad \forall m$. Therefore, processor $\tilde{m}_{k+1}^{k+1}$ makes the right decision for order $k + 1$, and we have: $\alpha_{k+1}^{k+1} = a_{k+1}^{k+1,\tilde{m}_{k+1}} = a_{k+1}^{\text{seq}}$. This completes the induction. □

## A.2. Proof of Theorem 3.2

We will use the following lemma, which will facilitate greatly the proof of Theorem 3.2.

**Lemma A.1.** *Let $Q_t := \{j \in [J] \mid \tau_j \le t\}$ be the set of nodes that run out of capacity before $t$ in the sequential scenario. We have, at any iteration $k \ge 1$, any time step $t \in [T]$, any product $i \in [I]$ and any process $m \in [M]$*

$$x_{t,i,j}^{k,m} \ge x_{t,i,j}^{\text{seq}}, \forall j \notin Q_t. \tag{1}$$

$$c_{t,j}^{k,m} \ge c_{t,j}^{\text{seq}}, \forall j \notin Q_t. \tag{2}$$

**Proof of Theorem 3.2.** We will invoke Lemma A.1 for the proof. To begin, we state that for all $k \ge 1$, the following holds

$$\alpha_t^k = a_t^{\text{seq}}, 1 \le t \le \tilde{\tau}_k. \tag{3}$$

We can see that (3) directly implies Theorem 3.2. We proceed to prove (3) by induction, assuming it holds when $1 \le k < k_c$. We want to prove the case for $k = k_c$. By induction, we have $x_{\tilde{\tau}_{k_c}-1,i,j}^{k_c,m} = x_{\tilde{\tau}_{k_c}-1,i,j}^{\text{seq}}$ and $c_{\tilde{\tau}_{k_c}-1,j}^{k_c,m} = c_{\tilde{\tau}_{k_c}-1,j}^{\text{seq}}$ for all $i \in [I], j \in [J]$ and $m \in [M]$. In particular, the set of nodes $\{j \in [J], c_{\tilde{\tau}_{k_c}-1,j}^{k_c,m} = 0\} = \{j \in [J], c_{\tilde{\tau}_{k_c}-1,j}^{\text{seq}} = 0\} = Q_{\tilde{\tau}_{k_c}}$ have run out of capacity at time $\tilde{\tau}_{k_c-1}$ for both the local process $m$ and the sequential scenario.

Now, consider $t \in [\tilde{\tau}_{k_c-1} + 1, \tilde{\tau}_{k_c})$, we have that in the sequential scenario no node is running out of capacity during this period (note that the next depleted one is at $\tilde{\tau}_{k_c}$). Therefore $Q_t = Q_{\tilde{\tau}_{k_c}}$ and $c_{t,j}^{\text{seq}} > 0, \forall j \notin Q_{\tilde{\tau}_{k_c}}$. Then by Lemma A.1, we have that for $t \in [\tilde{\tau}_{k_c-1} + 1, \tilde{\tau}_{k_c})$,

$$c_{t,j}^{k_c,m} \ge c_{t,j}^{\text{seq}} > 0, \forall j \notin Q_{t_{k_c}}.$$

Therefore, for $t \in [\tilde{\tau}_{k_c-1} + 1, \tilde{\tau}_{k_c})$, the process $m$ maintains the same feasible set of fulfillment nodes as the sequential

scenario:

$$\{\mathbf{1}(c_{t,j}^{k_c,m} > 0)\}_{j \in [J]} = \{\mathbf{1}(c_{t,j}^{\text{seq}} > 0)\}_{j \in [J]}.$$

Now consider an arbitrary product $i_c \in [I]$ and its orders $\mathcal{A}$ within the time period $[\tilde{\tau}_{k_c-1}+1, \tilde{\tau}_{k_c}]$: $\mathcal{A} \subset [\tilde{\tau}_{k_c-1}+1, \tilde{\tau}_{k_c}]$. The orders in $\mathcal{A}$ is processed sequentially by a process and for any $t \in \mathcal{A}$, we aim to show

$$\begin{aligned}
\alpha_t^{k_c} &= a_t^{k_c,m} \\
&= \pi(\{x_{t-1,i_c,j}^{k,m}\}_{j \in [J]}, \{\mathbf{1}(c_{t-1,j}^{k,m} > 0)\}_{j \in [J]}, \omega_t) \\
&= \pi(\{x_{t-1,i_c,j}^{\text{seq}}\}_{j \in [J]}, \{\mathbf{1}(c_{t-1,j}^{\text{seq}} > 0)\}_{j \in [J]}, \omega_t) \\
&= a_t^{\text{seq}}
\end{aligned}$$

This is evident since the capacity-feasible set remains identical between the local process and the sequential process, and the inventory levels $x_{t,i_c,j}^{k,m}$ and $x_{t,i_c,j}^{\text{seq}}$ share the same initial points at time $\tilde{\tau}_{k_c-1}$ and solely determined by the fulfillment decisions made for product $i_c$ at $t \in \mathcal{A}$. Thus all states and decisions remain the same between the local process and the sequential process during $[\tilde{\tau}_{k_c-1} + 1, \tilde{\tau}_{k_c}]$. This completes the proof. □

### A.2.1. PROOF OF LEMMA A.1

Note that the capacity consumed at a node is the sum of the inventories consumed at that node. Therefore, for given $j \in [J], t \in [T], x_{t,i,j}^{k,m} \ge x_{t,i,j}^{\text{seq}}$ for all $i \in [I]$ implies that $c_{t,j}^{k,m} \ge c_{t,j}^{\text{seq}}$. Thus it is sufficient to show (1).

We prove this by induction. (1) holds for $k = 0$ since all orders are not fulfilled and all inventories are not consumed. For any given $k_c \in \mathbb{N}, t_c \in [T]$, assume this holds for any $0 \le k < k_c, t \in [T]$ and $k = k_c, 1 \le t < t_c$. Now we want to prove (1) for $k = k_c, t = t_c$.

Consider a process $m$ associated with product subset $\mathcal{A}_m$ and order subset $\mathcal{P}_m$. Let the product at Time $t_c$ be $i_c$. It is sufficient to show

$$x_{t_c,i_c,j}^{k,m} \ge x_{t_c,i_c,j}^{\text{seq}}, \forall j \notin Q_{t_c} \tag{4}$$

since the inventory levels for other products will remain the same in processing order $t_c$.

If $i_c$ is handled by a process $m'$ that is not $m$, i.e., $i_c \in \mathcal{A}_{m'} \ne \mathcal{A}_m$, $m$ will skip the invocation of $\pi$ and take $a_{t_c}^{k,m} = a_{t_c}^{k-1,m'}$. In fact, this skip will occur for every order related to $i_c$, i.e., $a_t^{k,m} = a_t^{k-1,m'}$ for all $t \in [T]$ with $i(\omega_t) = i_c$. This implies that the inventory level for $i_c$ at process $m$ is the same as the inventory level at $m'$ from the previous iteration, thus (4) holds by induction:

$$x_{t_c,i_c,j}^{k,m} = x_{t_c,i_c,j}^{k-1,m'} \ge x_{t_c,i_c,j}^{\text{seq}}, \forall j \notin Q_{t_c}.$$

Next we consider the case $i_c$ is handled by process $m$, i.e., $i_c \in \mathcal{A}_m$. Recall that the dynamics of the states are

$$x_{t_c,i_c,j}^{k,m} = x_{t_c-1,i_c,j}^{k,m} - \mathbf{1}(a_{t_c}^{k,m} = j)$$
$$c_{t_c,j}^{k,m} = c_{t_c-1,j}^{k,m} - \mathbf{1}(a_{t_c}^{k,m} = j).$$

We aim to show that even the decision $a_{t_c}^{k,m}$ is wrong, meaning not the same as $a_{t_c}^{\text{seq}}$, the inventory monotonicity in (4) still hold. Specifically, we will show

$$a_{t_c}^{k,m} \in \{a_{t_c}^{\text{seq}}\} \cup Q_{t_c} \cup \{j \notin Q_{t_c}, x_{t_c-1,i_c,j}^{k,m} > x_{t_c-1,i_c,j}^{\text{seq}}\}. \tag{5}$$

(5) implies that when $a_{t_c}^{k,m} \neq a_{t_c}^{\text{seq}}$, we have either $a_{t_c}^{k,m} \in Q_{t_c}$, which is excluded when examing (4), or $a_{t_c}^{k,m} \in \{j \notin Q_{t_c}, x_{t_c-1,i_c,j}^{k,m} > x_{t_c-1,i_c,j}^{\text{seq}}\}$, which will maintain the inventory monotonicity after the state update. Thus, (5) implies (4).

To prove that (5) holds, we examine the capacity level at time $t_c$. Note that the capacity consumed is the aggregation of inventories consumed at any time point, thus for all $j \notin Q_{t_c}$,

$$
\begin{aligned}
c_{t_c-1,j}^{k,m} &= c_{0,j} - \sum_{i \in [I]} (x_{0,i,j} - x_{t_c-1,i,j}^{k,m}) \\
&\geq c_{0,j} - \sum_{i \in [I]} (x_{0,i,j} - x_{t_c-1,i,j}^{\text{seq}}) \quad \text{by induction} \\
&= c_{t_c-1,j}^{\text{seq}}.
\end{aligned}
$$

Thus, both capacity and inventory for the node $j \notin Q_{t_c}$ satisfy the monotonicity comparing to the sequential scenario:

$$c_{t_c-1,j}^{k,m} \geq c_{t_c-1,j}^{\text{seq}} \tag{6}$$
$$x_{t_c-1,i,j}^{k,m} \geq x_{t_c-1,i,j}^{\text{seq}} \tag{7}$$

We are going to use (6), (7), Assumptions 1-3 to show (5). Note that following our requirements for $\pi$ given by Assumption 3

$$\pi(\{x_{t_c-1,i_c,j}^{\text{seq}}\}_{j\in[J]}, \{\mathbf{1}(c_{t_c-1,j}^{\text{seq}} > 0)\}_{j\in[J]}, \omega_{t_c}) = a_{t_c}^{\text{seq}} \tag{8}$$

By Assumption 2, we can replace the inventory and capacity of the nodes in the set $Q_{t_c}$ by $x_{t_c-1,i_c,j}^{k,m}, c_{t_c-1,j}^{k,m}$ and obtain

$$
\begin{aligned}
\pi\Big( &\{x_{t_c-1,i_c,j}^{\text{seq}}\}_{j\notin Q_{t_c}} \cup \{x_{t_c-1,i_c,j}^{k,m}\}_{j\in Q_{t_c}}, \tag{9} \\
&\{\mathbf{1}(c_{t_c-1,j}^{\text{seq}} > 0)\}_{j\notin Q_{t_c}} \cup \{\mathbf{1}(c_{t_c-1,j}^{k,m} > 0)\}_{j\in Q_{t_c}}, \omega_{t_c}\Big) \\
&\in \{a_{t_c}^{\text{seq}}\} \cup Q_{t_c}
\end{aligned}
$$

Note that $c_{t_c-1,j}^{\text{seq}} > 0$ for $j \notin Q_{t_c}$ by the definition of $Q_{t_c}$ (the set of nodes running out of capacity before $t_c$, i.e.,

$Q_{t_c} = \{j | c_{t_c-1,j}^{\text{seq}} = 0\}$). Together this with (6), we have $\mathbf{1}(c_{t_c-1,j}^{k,m} > 0) = \mathbf{1}(c_{t_c-1,j}^{k,m} > 0)$ for $j \notin Q_{t_c}$. Thus (10) can be simplified to

$$
\begin{aligned}
\pi\Big( &\{x_{t_c-1,i_c,j}^{\text{seq}}\}_{j\notin Q_{t_c}} \cup \{x_{t_c-1,i_c,j}^{k,m}\}_{j\in Q_{t_c}}, \tag{10} \\
&\{\mathbf{1}(c_{t_c-1,j}^{k,m} > 0)\}_{j\in[J]}, \omega_{t_c}\Big) \\
&\in \{a_{t_c}^{\text{seq}}\} \cup Q_{t_c}
\end{aligned}
$$

Finally, by invoking Assumption 2-3 on the set of $\{j \notin Q_{t_c}, x_{t_c-1,i_c,j}^{k,m} > x_{t_c-1,i_c,j}^{\text{seq}}\}$ and replace $x_{t_c-1,i_c,j}^{\text{seq}}$ by $x_{t_c-1,i_c,j}^{k,m}$, one can verify that

$$
\begin{aligned}
a_{t_c}^{k,m} &:= \pi(\{x_{t_c-1,i_c,j}^{k,m}\}_{j\in[J]}, \{\mathbf{1}(c_{t_c-1,j}^{k,m} > 0)\}_{j\in[J]}, \omega_{t_c}) \\
&\in \{a_{t_c}^{\text{seq}}\} \cup Q_{t_c} \cup \{j \notin Q_{t_c}, x_{t_c-1,i_c,j}^{k,m} > x_{t_c-1,i_c,j}^{\text{seq}}\}.
\end{aligned}
$$

Thus (5) holds and this completes the proof. □

### A.3. Details of Problem Setup and Data-Generation Process

Our experiments use synthetic data based broadly on fulfillment-network and demand-distribution patterns observed at modern (moderately) large industrial-scale retailers. We consider a fulfillment network comprising $J = 30$ nodes located in the 30 most-populous states (each one in the most-populous city of the corresponding state). Although the distribution of demand among products was varied in certain experiments, in all cases, we provided sufficient inventory and fulfillment capacity at the network level (i.e. across all $J$ nodes) to permit the fulfillment of 80% of demand (not accounting for occasional cases of 'stranded' inventory, in which one is unable to use inventory left over at a node with no remaining capacity). For each product, $i$, the network inventory inventory was distributed across nodes *pro rata* based on population: $x_{0,i,j} \propto \text{population}_j$; similarly, network capacity was distributed *pro rata* across nodes based on population: $c_{0,j} \propto \text{population}_j$. For each (order, node) pair $(t, j)$, we computed the distance between the zip codes, $d_{tj}$, and then defined a reward for fulfilling Order $t$ from Node $j$ as follows:

$$r_j(\omega_t) = \frac{\max_j d_{tj} - d_{tj}}{\max_j d_{tj}}. \tag{11}$$

Finally, $a^\phi$ (the always-feasible action) was associated with a zero reward and had ties broken against it to ensure that we only choose it in the absence of any feasible alternatives.

We implement two practical optimizations that improve performance (but do not impact our theoretical analysis): First, if $t_{\text{reset}}$ is the smallest $t$ for which $\alpha_t^k \neq \alpha_t^{k-1}$, then we know that the actions evaluated for times $t < t_{\text{reset}}$ are correct and it is sufficient to start the $k$th Picard iteration at time $t = t_{\text{reset}}$. Second, as opposed to running

Picard iteration over the entire horizon, we run the iteration in 'chunks' of size max_steps and move on to the next chunk only after convergence of the preceding one. More precisely, we run the **for** loop in Line 5 of the algorithm over $t \in [t_{\text{reset}}, \min(T, t_{\text{reset}} + \text{max\_steps})]$. Tuning the max_steps parameter thus trades off the need for synchronization (the number of iterations of the **while** loop in Line 2), with the potential for 'wasting' computation (the number of iterations of the **for** loop in Line 5).

We consider a high-level (JAX (Bradbury et al., 2018)) implementation of Picard that does not require custom kernels. Notably, Line 3 of our algorithm is implemented via jax.vmap so that we forego fine-grained control of batching evaluations of $\pi(\cdot)$.

All experiments were run on a single A100 GPU with 40GB of VRAM, at 16 bit precision. All key operations are jit-compiled. We also implement the sequential baseline on GPU, where it can leverage parallelism in subroutines such as matrix multiplication.

### A.4. Practical performance with capacity-dependent policies

The bound on number of iterations until convergence in Section 3.2 assumes that the fulfillment policy does not depend on node capacity. As a final robustness check, however, we are interested in gauging the empirical performance of Picard Iteration for policies that *do* depend on capacity (beyond ensuring feasibility), a regime for which our theory does not directly apply. We therefore conduct an ablation study using a simplified version of the policy that penalizes capacity usage at low-capacity nodes using shadow prices, with strength modulated by a parameter $\gamma$. Ranging $\gamma$ from 0 to $\infty$ interpolates between a closest-node policy and a policy that always fulfills from the node with greatest remaining capacity (an unrealistic extreme scenario); for reference, a value of $\gamma = 1.0$ allows the node with greatest remaining capacity to improve its ranking by up to $J - 1$ positions (i.e. even the most-expensive node may be chosen if its remaining capacity is sufficiently attractive). The results are shown in Table 3 and demonstrate that empirically, the constraint on capacity (non-)dependence is non-vacuous but can be relaxed substantially: Performance is virtually unchanged for realistic settings of $\gamma$ that incorporate shadow costs on capacity.

*Table 3.* Ablation study on a policy that discounts node proximity by (remaining) capacity scarcity

| $\gamma$ | Conflicts | Speedup |
|---|---|---|
| 0 (no capacity dependence) | 13 | 441x |
| 0.5 | 14 | 427x |
| 1.0 | 15 | 401x |

### A.5. Effect of initial action sequence

Loosely speaking, the closer the initialization is to the correct trajectory, the fewer iterations will be required. We test the sensitivity of Picard Iteration to the initialization by experimenting with three natural initializations:

1. Unfulfill (441x Speedup): initially, all orders are unfulfilled. This setting corresponds to the theoretical results and experiments in the main body of the manuscript.

2. Naive (358x Speedup): Initially, fulfill all orders from their nearest fulfillment center, ignoring inventory and capacity constraints.

3. Random (390x Speedup): Initially, fulfill all orders from a random FC.

Clearly, different initialization methods have an impact on the simulation speed, although massive speedups are possible regardless. Here, it appears that the "Unfulfill" strategy works the best. The intuitive explanation for this is consistent with our analysis: using this initialization strategy, after the first iteration, each processor will have correctly accounted for inventory and capacity consumed by orders for that processor's own products – in some sense, a "greedy" initialization which is less naive than the "Naive" approach above. Further, unlike the alternatives, when initializing each chunk, this procedure is able to account for inventory and capacity consumed in previous chunks.

### A.6. Limitations and worst-case set-up

To understand potential limitations, consider a back-of-the-envelope analysis of the key factors contributing to the simulation runtime:

- $t_\pi$, the time to execute the policy at a single time step

- $t_f$, the time to execute one step of the dynamics

- $T$, the problem horizon (number of steps)

- $K$, the number of iterations required to converge

- $M$, the number of parallel processors available

Sequential execution requires time $T(t_\pi + t_f)$, whereas Picard iteration requires:

$$K \left( \frac{T t_\pi}{M} + T t_f \right) \quad \text{(for } M \leq T\text{)}$$

Potential limitations are easy to see from this equation:

- If $K$ is sufficiently large, this can offset the benefits of parallelization. As we show in the paper, $K$ is provably small in many useful cases, but this does not hold universally; see the worst-case setup in response to the next question.

- If $t_f$ is sufficiently large (i.e., dynamics are very expensive), then the term $KTt_f$ can dominate. In many real problems (such as the FO and inventory management settings we analyze), the dynamics are trivial to compute, so $t_f$ is negligible. For settings where this is not true, this dependence can be improved by computing expensive parts of the state transitions in parallel and caching them, as we do for actions.

One can construct a problem and Picard iteration scheme (i.e., allocation of tasks to processors), for which convergence requires $T$ iterations, although this is quite pathological. Consider a setting with a single product, two parallel processors and two fulfillment centers (FC), FC1 and FC2, starting with equal capacity. Suppose that event assignments alternate between the two processors (i.e., processor 1 is responsible for odd-numbered events, and processor 2 for even-numbered events). Suppose that the policy is simply to fulfill orders from the FC having the largest remaining capacity, with a preference for FC1 over FC2 in case of ties. Finally, suppose that the initial trajectory provided to Picard Iteration is to unfulfill all orders. In this scenario, at the first iteration, both processors will fulfill all orders from FC1; In the second iteration, starting from the time $t = 2$, both processors fulfill all subsequent orders from FC2; and in the third, starting from the time $t = 3$, both processors fulfill all subsequent orders from FC1. They continue to alternate like this for $T$ iterations until convergence. Note that, in the implementation of Picard iteration that we analyze, since there is only one product, all events would be assigned to the same processor, which avoids these conflicts.

## B. Extension: Inventory Control with Replenishment

### B.1. Problem formulation

We describe here the *One Warehouse Multi-Retailer* (OWMR) problem. We have one central *distribution center* and $N$ *retailers*, indexed by $n \in [N]$. Let $x_{t,0} \in \mathbb{R}$ be the inventory of the central warehouse at time $t$ and $x_{t,n} \in \mathbb{R}$ be the inventory level of retailer $n$ at time $t$. The state at time $t$ is then

$$s_t := (x_{t,0}, x_{t,1}, \ldots, x_{t,N}).$$

For now, we assume all replenishments are immediate, hence we have no in-transit inventory. Each period $t$, a *demand* arrives at retailer $n(\omega_t)$ of quantity $d(\omega_t)$. A policy

$\pi$ decides (i) how much the central warehouse replenishes, $q_{t,0}$, and (ii) how much retailer $n(\omega_t)$ replenishes from the central warehouse, $q_{t,n(\omega_t)}$:

$$a_t := \pi(s_t, \omega_t) = (q_{t,0}, q_{t,n(\omega_t)}) \in \mathbb{R}^2.$$

The inventories evolve as follows:

$$x_{t+1,n} = \begin{cases} (x_{t,n(\omega_t)} - d(\omega_t))^+ \\ \quad + \min(q_{t,n(\omega_t)}, x_{t,0}), & \text{if } n = n(\omega_t), \\ x_{t,n}, & \text{if } n \neq n(\omega_t). \end{cases}$$
(12)

The inventory of the central warehouse evolves as:

$$x_{t+1,0} = (x_{t,0} - q_{t,n(\omega_t)})^+ + q_{t,0},$$

where $(\cdot)^+$ represents the non-negative part. We also assume there is an upper bound for how much inventory can be hold at each location: $x_{t,n} \leq c_n \, \forall n = 0, 1, \ldots, N$.

### B.2. Theoretical Guarantee

**Policy.** We consider the following policy $\pi$. For each retailer $n \in [N]$, the replenishment decision $q_{t,n} = \pi_n(x_{t,n}, \omega_t)$ is a function of its local inventory and the exogenous information (*this exogenous information can include the historical demand across retailers*). The function $\pi_n$ can be arbitrary (e.g., a neural net) except satisfying the capacity condition $(x_{t,n} - d(\omega_t))^+ + q_{t,n} \leq c_n$ and positivity condition $q_{t,n} \geq 0$. For the central warehouse, we consider an $(s, S)$-policy (i.e., an order is placed to increase the item's inventory position to the level $S$ as soon as this inventory position reaches or drops below the level $s$). That is

$$x_{t+1,0} = \begin{cases} S & x_{t,0} - q_{t,n(\omega_t)} \leq s, \\ x_{t,0} - q_{t,n(\omega_t)} & \text{otherwise.} \end{cases}$$
(13)

The $(s, S)$-policy is known to be optimal in a single-product setting with fixed ordering costs (Scarf et al., 1960), making it a common starting point for practical inventory management. We further assume $s \geq c_n, \forall n$ to simplify the analysis, whereas the experiments in Appendix B.3 are performed without this constraint.

**Picard Iteration.** We allocate time steps based on retailers: $T_n = \{t \mid n(\omega_t) = n\}$, with the intuition that the replenishment decisions across retailers are weakly entangled only through the central warehouse. Under this setup, the convergence speed of Picard iteration can be bounded by $R$, the number of times the central warehouse is replenished:

$$R := \sum_{t=0}^{T} 1(q_{t,0}^{\text{seq}} > 0).$$

In many practical scenarios, it is reasonable to expect that $R \ll T$ (e.g., when the cost of replenishing the central warehouse is high). In particular, we establish the following theorem:

**Theorem B.1.** *For the OWMR problem and the corresponding policy $\pi$ described in this section, the Picard Iteration converges in at most $2R$ iterations.*

**Proof Sketch.** Under the $(s, S)$-policy, we have $\forall t, x_{t,0} \geq s \geq q_{t,n}$. Thus, the replenishment decisions at local retailers are completely separable (i.e., $q_{t,n}^{k,m} = q_{t,n}^{\text{seq}}$), making it sufficient to consider the decisions for the central warehouse. The dynamics of the central warehouse at each processor evolve as follows: when $n(\omega_t) \notin \mathcal{N}_m$, where $\mathcal{N}_m$ denotes all retailers handled by processor $m$, we have

$$
x_{t,0}^{k,m} = \begin{cases} (x_{t-1,0}^{k,m} - q_{t,n(\omega_t)}^{\text{seq}})^+ & \text{if } q_{t,0}^{k-1} = 0, \\ S & \text{otherwise.} \end{cases}
$$

If the decision to replenish the central warehouse at iteration $k - 1$ is incorrect, then $x_{t,0}^{k,m}$ will be wrongly replenished to $S$, causing error propagation and presenting a major obstacle to proving convergence.

Fortunately, the following lemma ensures the correctness of the central warehouse decisions, leading to the guarantees stated in the theorem:

**Lemma B.2.** *Suppose $q_{t_1,0}^{\text{seq}} > 0$ for some $t_1$. Let $t_2 = \min\{t \mid t > t_1, q_{t,0}^{\text{seq}} > 0\} \cup \{T\}$ be the next time step when the central warehouse is replenished. Assume $q_{t,n}^{k,m} = q_{t,n}^{\text{seq}}$ for all $t \in [T], n \in [N]$ and $q_{t,0}^{k,m} = q_{t,0}^{\text{seq}}$ for $t \leq t_1$. Then for any $\{q_{t,0}^k, \forall t \in (t_1, t_2]\}$ at the $k$th iteration, after 2 iterations, we have $q_{t,0}^{k+2} = q_{t,0}^{\text{seq}}, \forall t \in (t_1, t_2]$.*

The intuition for Lemma B.2 is that, regardless of how incorrect $q_{t,0}^k$ is at the $k$th iteration, the $k + 1$th iteration will not replenish the central warehouse between $t \in (t_1, t_2)$ because the incorrect decisions from other processors will only delay the time that the inventory drops below $s$. Then, the $k + 2$th iteration will ensure the central warehouse is replenished at $t_2$. Details are omitted for simplicity.

### B.3. Experiments

**Base case.** We consider a base setting inspired by Walmart, which operates over 4,000 stores (Walmart, 2022) to fulfill their online demand. We simulate one year of data of one product that sells about 10,000 units per day across the 4,000 stores (2.5 units per store per day). We set the replenishment policy of the central warehouse and local stores such that the central warehouse replenishes about once weekly. Such frequency is often observed in practice due to the presence of fixed order costs. Simulating one year of data ($T = 10,000 \times 365 = 3,650,000$) yields an empirical speed-up

of **230x** compared to a naive sequential evaluation. We use 4,000 parallel workers and set max_steps=130; the policy call consists of a Multi-Layer Perceptron with two hidden layers with width 512. As per Theorem B.1, Picard Iteration needs 110 iterations (twice per weekly replenishment of the central warehouse). Next, we demonstrate the robustness of our results through an ablation study, systematically altering various problem and algorithm parameters.

**Ablation Study.** We start by evaluating robustness under heavy-tailed demand. To do this, we define the weights for store selection as $w_i = (1 - \alpha)^i$, where $i$ represents the store index and $\alpha \in [0, 1]$ controls the skewness of the distribution. The weights are normalized into probabilities $p_i = w_i / \sum_j w_j$. When $\alpha = 0$, the distribution is uniform across stores. As $\alpha$ increases, the distribution becomes progressively more skewed, favoring stores with lower indices. For instance, 20% of retailers are responsible for 80% of the demand when $\alpha = 0.002$. Table 4 presents the speedup of Picard iteration compared to sequential evaluation. As expected, the performance improvement decreases as the demand distribution becomes more heavy-tailed. Note that we assign one processor per retailer; further optimization of the partitioning could yield better results, making the reported speedups a conservative estimate.

| $\alpha$ | **Speed-up** | $\alpha$ | **Speed-up** |
|---|---|---|---|
| 0.000 | 230x | 0.006 | 55x |
| 0.001 | 227x | 0.007 | 50x |
| 0.002 | 113x | 0.008 | 44x |
| 0.003 | 111x | 0.009 | 39x |
| 0.004 | 74x | 0.010 | 36x |
| 0.005 | 72x | 0.011 | 33x |

*Table 4.* Speedup of Picard Iteration relative to sequential, for different $\alpha$ values in the demand distribution.

Figure 4 illustrates the effect of varying the number of retailers $N$ and batch size on speedup. As the number of retailers increases, we observe speedups reaching up to **380x** for 1,000 retailers, followed by a gradual decline for larger counts, likely due to GPU memory. Varying the batch size for Picard iteration exhibits a comparable trend, with speedups peaking at **383x** for 1,000 parallel workers before slightly decreasing at larger batch sizes.

Lastly, we modify the width of the two hidden layers in our DNN. While our speedups remain high, they gradually decline as the neural network size increases.

## C. Extension: Fulfillment Problem with Refreshing Capacity

Consider a setting where there are exogenous steps that periodically refill the capacity with a fixed amount, modeling, for example, the daily or hourly replenishment of a fulfillment center's capacity.

**Theorem C.1.** *For $T$ steps, assume there are $K$ refills, the frequency of each product in the demand is bounded by $M$, and the number of processors is $P$. Then, the computational complexity of the Picard iteration is given by (assume $T \geq P$)*

$$O\left(\frac{T(K+J)M}{P}\right),$$

*where $J$ is the number of fulfillment nodes.*

*Proof.* We model each refill as introducing a new fulfillment node into the system. By treating these nodes as part of the overall network, we can then apply the reduction to obtain the bound. $\square$

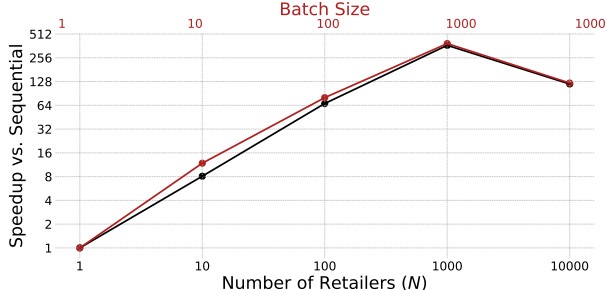

*Figure 4.* Speedup of Picard Iteration relative to sequential, as a function of different number of batch size and retailers $N$.

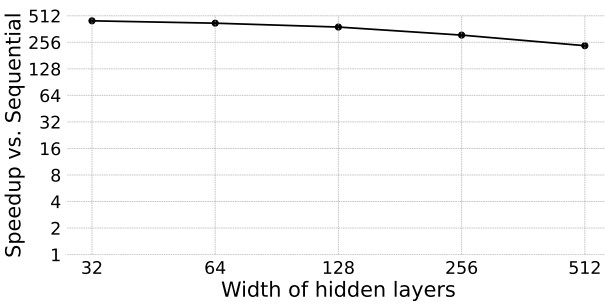

*Figure 5.* Speedup of Picard Iteration relative to sequential, as a function of size of neural net.

