# OpenReview forum: "Speeding up Policy Simulation in Supply Chain RL"
_ICML.cc/2025/Conference — ICML 2025 poster_

### Official Review · Reviewer_4umB · 2025-02-18

**Overall Recommendation:** 4

**Summary:**

This paper provides an iterative algorithm that is easy to parallelize with GPUs to speed up policy evaluation in RL. The theoretical analysis is specific to supply chain optimization, where the authors leverage the assumption that demand is significantly higher than supply so that actions at different time steps are largely independent, proving that the algorithm converges in a small number of iterations (independent of the time horizon length). The computational studies provide empirical evidence that largely aligns with the theoretical analysis and extends the applications to RL settings beyond supply chain analysis.

**Claims And Evidence:**

The claims are generally well-supported. I only have one concern related to the introduction.

While the introduction outlines a general challenge in supply chain optimization (SCO), the theoretical analysis focuses specifically on fulfillment optimization (FO) problems that are not supply-constrained and have no replenishment. The assumptions made in the theoretical analysis may not generalize well beyond FO problems. I suggest revising the introduction to more clearly state these key assumptions and the specific focus of the paper to better align with its scope and contributions.

**Essential References Not Discussed:**

I didn't identify any missing references that are essential.

**Experimental Designs Or Analyses:**

Yes, I checked. I didn't identify any major issues. Some suggestions are mentioned in the "strengths and weaknesses" section below.

**Methods And Evaluation Criteria:**

The proposed method focuses on letting the algorithm converge in a small number of iteration. The numerical studies report the number of iterations taken to converge. They are well aligned.

**Other Comments Or Suggestions:**

- PO is not defined in the main body.

**Other Strengths And Weaknesses:**

**Strengths**:

- Overall, the paper is very well-written and thus easy to follow.
- The idea aligns with the recent trend of taking advantage of GPUs to accelerate optimization algorithms that are predominantly implemented with CPUs to date, thus could be interesting to a broad community.
- The numerical studies present significant speedups compared to existing methods.

**Weaknesses**:

- **Exposition**. My primary concern relates to the exposition, specifically the following two points.
    -  **Intro**. While the introduction outlines a general challenge in supply chain optimization (SCO), the theoretical analysis focuses specifically on fulfillment optimization (FO) problems that are not supply-constrained and have no replenishment. The assumptions made in the theoretical analysis may not generalize well beyond FO problems. I suggest revising the introduction to more clearly state these key assumptions and the specific focus of the paper to better align with its scope and contributions.
    - **Section 3**. In section 3, the authors choose to present a detailed proof of a specific result (Theorem 3.1) and defer the proof of a general result (Theorem 3.2) to the appendix. Given that presenting Theorem 3.1 does not seem to yield much additional insight, the authors might consider restructuring this section to highlight the most significant contribution, i.e., Theorem 3.2, and present Theorem 3.1 as a corollary. Alternatively, the authors might restructure the discussions around Theorem 3.1 to highlight the insights obtained by analyzing this simple case---which might be hard to obtain given the complexity of the more general case.

- **Theorem 3.2**. $\mathcal{Q}_T$ was defined as a set in lines 262--263, but is used as a scalar in this statement.

- **No replenishment**. While the assumption of no replenishment enables elegant theoretical analysis, it warrants more careful justification since it may deviate from common supply chain practices.

- **Data generation**. To ensure that the paper is self-contained, it would be helpful if the authors provided a brief description of their data generation process.

**Questions For Authors:**

- Section 4.1. What do you use an MLP to approximate the greedy algorithm instead of implementing it exactly?
- Section 4.1. In line 320, what does "increased conflicts" mean?

**Relation To Broader Scientific Literature:**

The idea aligns with the recent trend of taking advantage of GPUs to accelerate optimization algorithms that are predominantly implemented with CPUs to date, thus could be interesting to a broad community.

**Theoretical Claims:**

I reviewed the proofs of Theorems 3.1 and 3.2. I didn't spot any major issues.

---

> ### Author Rebuttal · Authors · 2025-04-01
>
> Thank you for your detailed review! Please see our responses below.
>
>
> ## Comments on Exposition
> Thank you for these helpful comments! We will be sure to clarify these points in the updated manuscript. Please see specific clarifications below.
>
> ### Re: Intro and scope
>
> You are correct that, while Picard Iteration applies immediately to any MDP, the theoretical analysis of convergence is problem specific. However, there is actually a variety of problems in SCO that can be studied using similar methods. Our analysis for FO problem can admit replenishment as long as the replenishing frequency is not too high (Appendix C). Beyond the FO problem, we also address the inventory control problem — another representative example of SCO — in the Appendix B (due to space constraints). We develop a theoretical analysis for the inventory control problem, and demonstrate demonstrate significant speedups in experiments.
>
>   That said, your suggestion regarding exposition is well taken—we will revise the introduction in the final version to better clarify the scope and focus of the paper, as well as to clearly state our theoretical assumptions.
>
> ### Re: Exposition of Theorem 3.1 / 3.2
>
> Thank you for this suggestion! We will make the exposition clearer in this section. To clarify the goals of the current manuscript:
>
> Theorem 3.1 is an informal statement of Theorem 2.2, with the goal of enabling the reader to quickly grasp the main idea of our result: that Picard Iteration provides a factor $\sim M/J$ speedup, which we obtain by bounding the number of iterations as $|\mathcal{Q}_{T}|$.
>
>
>   In Section 3.1 we then prove a special case of the main theorem (Theorem 3.1/3.2) for a specific greedy policy, which is easier to analyze. The goal of this section is to demonstrate the more general proof architecture, and to provide intuition for why one might hope that the number of iterations required should be small: The action taken at every time step is either going to be correct or will go to a node that would run out of capacity in the sequential scenario (Lemma 3.2).  Theorem 3.1/3.2 follows directly from this Lemma, for a greedy policy.
>
>   Subsequently, Section 3.2 states that the same result holds for a broader class of policies. As the analysis is more involved (but the result is the same), we defer this analysis to the Appendix.
>
> ### Re: No replenishment
> Thanks for raising this point. First, we note that the setting without replenishment is commonly studied in the literature (e.g., https://pubsonline.informs.org/doi/10.1287/educ.2019.0199).  That said, adding replenishment indeed makes the setting more comprehensive and realistic. In Appendix B.3 we show the method's applicability in a one-warehouse-multi-retailer problem with replenishment. We find that the number of iterations is bounded by twice the number of times the central warehouse is replenished during the horizon, which is typically small compared to the total number of timesteps. We also confirm this result empirically, achieving up to 380x speed-ups for settings with 1,000 retailers. In addition, in Appendix C, we briefly present our generalization of the FO problem with replenishment, and show that the Picard iteration still converges quickly if the frequency of replenishment is low. We can add more details for the discussion in the revised version.
>
> ### Re: usage of $\mathcal{Q}_T$
>
> Thank you, we will correct this -- it should be the cardinality $|\mathcal{Q}_{T}|$.
>
> ## Specific Questions
> 1. *Re: MLP approximation to greedy:* The goal here is to simulate a computational workload representative of the kinds of heavy-weight neural network policies often used in reinforcement learning for supply chain. To this end, we need to experiment with some sort of neural network policy which behaves predictably, which we obtain by cloning the greedy policy. We will clarify this in the experimental setup.
> 2. *Re: "Increased conflicts"*: Thank you for noting this -- we mean that the number of required Picard iterations increases.  We will correct the language in this section to make it consistent with the rest of the paper.

---

> > ### Comment · Reviewer_4umB · 2025-04-04
> >
> > Thank you for your response. All my concerns have been properly addressed and I remain positive about this paper.

---

### Official Review · Reviewer_UC5Y · 2025-03-07

**Overall Recommendation:** 4

**Summary:**

The paper studies the acceleration of policy simulation for supply chain problems often solved via RL. The objective is to accelerate sequential evaluations using caching mechanisms to be able to parallelize/batch simulations. Numerical experiments are performed for supply chain problems and beyond.

**Claims And Evidence:**

The claims of the paper appear valid.

**Essential References Not Discussed:**

n.a.

**Experimental Designs Or Analyses:**

The experimental design is reasonable. Examples for supply chain problems as well as from MuJoCo are performed. Results are suitably discussed.

**Methods And Evaluation Criteria:**

The approach used is interesting. The presentation of the concepts is reasonable.

**Other Comments Or Suggestions:**

page 1 contains the claim: “For a policy parameterized by a non-trivial deep neural network (DNN), the task of simply simulating a single sample path under the policy can thus take several hours (or more) in the context of an SCO problem”

Do you have a supporting reference for the claim?

**Other Strengths And Weaknesses:**

Strengths:

- Analytical results (convergence)

- The provided experiments are overall convincing

Weaknesses:

- Limitations of the approach could be discussed in more detail

**Questions For Authors:**

(1) Do hyperparameters have to be tuned?

(2) What are the limitations/downsides of the approach?

(3) Is there a worst case setup?

(4) Can the approach be used outside Supply Chain problems? How do the assumptions have to be adapted/generalized?

**Relation To Broader Scientific Literature:**

The related literature is suitably discussed.

**Theoretical Claims:**

The theoretical claims appear solid. However, I did not check all proofs provided in the Appendix in full detail.

---

> ### Author Rebuttal · Authors · 2025-04-01
>
> Thank you for your detailed review! Please see our responses below.
>
> *Re: Do you have a supporting reference for the claim?*
>
> Our claim is based on direct experience from industry deployments and discussions with practitioners at major e-commerce companies. These conversations consistently confirmed the need for hours (if not days) of simulation time. It became very apparent in these calls with practitioners that the daily simulation need vastly exceeds the time available, such that often approximations (such as only simulating a portion of products and extrapolating their resource usage and cost to other products) are used to make daily decisions. Our work, however, may allow many of these simulations to be run in the available time window.  Unfortunately, we are not allowed to provide a specific citation for these useful discussions.
>
> Note that the fulfillment optimization problem only comprises a small piece of the daily supply chain puzzle to be solved. For instance, the outcome of our fulfilment simulation also informs resource planning at fulfilment centers, transportation planning, etc. It is thus essential the simulation happens fast, such that other algorithms can use the output.
>
> While confidentiality prevents us from explicitly citing these discussions, we've ensured our numerical settings are realistic and conservative, drawing on publicly available industry data from Amazon and Walmart, as cited at the beginning of Section 4.1 and in Appendix B.3. We believe our chosen settings are representative for a large amount of companies (and conservative for truly large-scale settings).
>
> ## Specific questions
>
> ### 1. Hyperparameter tuning
>
> Yes, the main hyperparameter to tune is what we have called "chunk size," i.e., the number of steps to execute at each iteration. While choosing a good value for this parameter can substantially improve performance, Picard Iteration still achieves massive speedups relative to sequential simulation for a wide range of chunk sizes -- our experiments show robust speedups of 200-440X across a range of chunk sizes from 30 to 300. Hyperparameters can be tuned at low cost by selecting those that maximize simulation speed for a small sub-trajectory.
>
> ### 2. Limitations
>
> To understand potential limitations, consider a back-of-the-envelope analysis of the key factors contributing to the simulation runtime:
>
>  - $t_{\pi}$, the time needed to execute the policy at a single time step
>  - $t_f$, the time needed to execute one step of the dynamics
>  - $T$, the problem horizon (number of steps)
>  - $K$, the number of iterations required for Picard iteration to converge
>  - $M$, the number of parallel processors available.
>
> Sequential execution requires time $T(t_\pi + t_f)$, whereas Picard iteration requires $K(Tt_\pi/M + T t_f)$ (for $M \leq T$). Potential limitations are easy to see from this equation:
>
> - If $K$ is sufficiently large, this can offset the benefits of parallelization. As we show in the paper, $K$ is provably small in many useful cases, but this does not hold universally; see the worst case setup in response to the next question.
> - If $t_f$ is sufficiently large (i.e., dynamics are very expensive) then the term $KTt_f$ can dominate. In many real problems (such as the FO and inventory management settings we analyze) the dynamics are trivial to compute, so $t_f$ is negligible. For settings where this is not true, this dependence can be improved by computing expensive parts of the state transitions in parallel and caching them, as we do for actions.
>
> ### 3. Worst-Case Setup
>
> Indeed, one can construct a problem and Picard iteration scheme (i.e., allocation of tasks to processors), for which convergence requires $T$ iterations, although this is quite pathological.
>
> Consider a setting with a single product, two parallel processors and two fulfillment centers (FC), FC1 and FC2, starting with equal capacity. Suppose that event assignments alternate between the two processors (i.e., processor 1 is responsible for odd-numbered events, and processor 2 for even-numbered events). Suppose that the policy is simply to fulfill orders from the FC having the largest remaining capacity, with a preference for FC1 over FC2 in case of ties. Finally, suppose that the initial trajectory provided to Picard Iteration is to unfulfill all orders.
>
> In this scenario, at the first iteration, both processors will fulfill all orders from FC1; In the second iteration, starting from the time $t=2$, both processors fulfill all subsequent orders from FC2; and in the third, starting from $t=3$, both processors fulfill all subsequent orders from FC1. They continue to alternate like this for $T$ iterations until convergence.
>
> Note that, in the implementation of Picard iteration that we analyze, since there is only one product, all events would be assigned to the same processor, which avoids these conflicts.

---

### Official Review · Reviewer_D3aU · 2025-03-13

**Overall Recommendation:** 4

**Summary:**

This paper introduces the Picard Iteration algorithm to accelerate policy simulation in reinforcement learning for supply chain optimization (SCO) problems, where simulating a single trajectory can take hours due to the serial nature of policy evaluations. The key innovation allows for the batched evaluation of a policy across a single trajectory by assigning policy evaluation tasks to independent processes and using a cached evaluation system that iteratively converges to the correct sequential simulation. The authors prove that for supply chain problems, this approach converges in a number of iterations independent of the time horizon, resulting in significant speedups. Empirical results demonstrate a 400x speedup on large-scale fulfillment optimization problems using a single GPU compared to sequential evaluation, and a 100x improvement over the Time Warp algorithm (a traditional parallel discrete event simulation approach). Additional experiments show that the method generalizes well to inventory control problems and has promising applications to standard reinforcement learning environments outside of supply chain, potentially providing speedups of up to 40x in MuJoCo environments.

**Claims And Evidence:**

Yes, the claims made in the paper are well-supported by both theoretical analysis and empirical evidence. The authors provide formal proofs for the convergence of their Picard Iteration algorithm in supply chain optimization contexts, along with extensive experimental results that demonstrate the claimed speedups across different problem settings, batch sizes, and demand distributions. The comparison against baseline methods like Time Warp is thorough, and they extend their evaluation to other domains like MuJoCo environments to show broader applicability.

**Essential References Not Discussed:**

N/A

**Experimental Designs Or Analyses:**

I checked the experimental designs and analyses, and I feel the current setup is good. The authors present comprehensive experiments across different problem settings, with appropriate controls and ablation studies that demonstrate the robustness of their approach.

While I think it could be possible to strengthen the paper by talking about how the initial actions affect the convergence - for example, explicitly showing that setting the actions to a better policy can speed up the convergence. In the MuJoCo experiments, they use the previous policy iterate to initialize the cache, which is intuitive, but a more systematic study of initialization strategies could provide additional insights.

Also, for the normalized error metrics, I think it would be interesting to show the RMSE of actions not only the states. This would provide a more complete picture of convergence behavior and could potentially reveal different convergence patterns between state trajectories and action trajectories.

**Methods And Evaluation Criteria:**

Yes, the proposed methods and evaluation criteria are well-suited for the problem. The authors evaluate their Picard Iteration algorithm on relevant supply chain optimization problems (fulfillment optimization and inventory control), using realistic parameters based on industry settings. Their benchmarking against sequential simulation and Time Warp provides appropriate baselines. The additional exploration in MuJoCo environments demonstrates broader applicability beyond the supply chain domain.

**Other Comments Or Suggestions:**

While this paper focuses on scenarios where policy evaluation is expensive compared to state transitions, there's also an opportunity to consider the inverse case. In many complex environments like physics simulations or large-scale agent models, state calculations can be more expensive than policy evaluations. Perhaps a "cached state" approach could be developed, mirroring the Picard Iteration but in reverse. Instead of caching actions, we could cache intermediate states and only recalculate states when necessary.

**Other Strengths And Weaknesses:**

Strengths

Practical Impact: The paper addresses a significant bottleneck in applying RL to large-scale supply chain problems. A 400x speedup could transform what were previously impractical approaches into viable solutions.

Theoretical Guarantees: The authors provide formal proofs that Picard converges to the correct trajectory and, more importantly, that it does so in a number of iterations independent of the simulation horizon length for SCO problems.

Comprehensive Evaluation: The experimental validation is extensive, testing across various problem configurations, including different batch sizes and demand distributions.

Weakness:

While the paper demonstrates impressive performance gains, it provides limited detail on implementation challenges. Although the authors mention using JAX and GPU acceleration, it would benefit the practitioners if they could talk more about optimization of synchronization between iterations, or parameter tuning strategies like selecting optimal chunk sizes.

**Questions For Authors:**

1 How does the choice of initial action sequence affect convergence speed in practice across different environments? Have you systematically studied how different initialization strategies impact convergence rates in various problem settings, and are there heuristics for selecting initial action sequences that could accelerate convergence?

2 In your convergence analysis for MuJoCo environments (Figure 3), you measure relative RMSE of state trajectories. Have you also analyzed the convergence behavior of action trajectories?

3 Figure 3 shows significantly different convergence patterns across MuJoCo environments, with some converging in just 2-3 iterations while others require 10+ iterations. Do you have insights into why these differences occur? Are there specific properties of these environments (state transitions, policy complexity, dynamics) that predict faster or slower convergence with Picard iteration?

**Relation To Broader Scientific Literature:**

The paper's key contribution of using Picard iteration for parallel policy simulation relates to several research areas. It addresses limitations in parallel discrete event simulation methods like Time Warp, which don't work well for supply chain problems lacking local causality properties. The approach complements existing parallel reinforcement learning literature (e.g., Asynchronous RL, RLlib) by focusing on accelerating a single trajectory rather than parallelizing across multiple trajectories or agents.

**Theoretical Claims:**

I only checked the claims in the main text and I think they are reasonable. The proof approach for Theorem 3.1 in the special case (greedy policy without inventory constraints) is logically sound and builds intuitively on the properties of wrong actions and capacity exhaustion. The setup of Lemma 3.2 and its application to prove convergence in J+1 iterations follows naturally.

---

> ### Author Rebuttal · Authors · 2025-04-01
>
> Thank you for your detailed review! Please see our responses below.
>
> ## Implementation Details
> *Re: "...it provides limited detail on implementation challenges..."*.
>
> Thank you for raising this point. Due to space constraints, we provide detailed implementation specifics in Appendix A.3. As you note, chunk size significantly affects performance; however, our experiments show robust speedups of approximately 200-440X across a range of chunk sizes from 30 to 300; we can provide further details in the Appendix.
>
> Selecting good hyperparameters is straightforward: a reasonable heuristic for tuning this hyperparameter is to simulate a sub-trajectory of length $\ll T$, and select the values which minimize the time required to simulate this sub-trajectory.
>
> We will provide these details in the manuscript, and we also plan to release a package implementing Picard iteration. The supplementary materials provided with our submission also contain all the code used to generate the experiments in the paper.
>
> ## Settings with expensive dynamics
>
> *Re: "While this paper focuses on scenarios where policy evaluation is expensive compared to state transitions"*
>
> This is a great point. Actually, to a large extent, what one considers to be "state" vs. "action" in our framework is a design choice. In other words, the user has flexibility to determine which parts of the combined vector $(s_{t+1}, a_t)$ are generated in parallel and cached (i.e., treated as the "action"), and which are recomputed on the fly by each worker (i.e., treated as the "state"). The most efficient partitioning of state and action is problem dependent, and as you note, could involve a reversal of the nominal roles if dynamics are expensive.
>
>
> ## Specific Questions
>
> ### 1. Re: effect of initial action sequence
>
> Loosely speaking, the closer the initialization is to the correct trajectory, the fewer iterations will be required. In a policy optimization setting, the most obvious and effective approach to initialization is to use trajectories collected using the previous policy iterate to initialize Picard Iteration, to collect trajectories using the current policy iterate. This is what we do in our policy optimization and MuJoCo experiments.
>
> In the policy evaluation setting for FO, we study the sensitivity of Picard Iteration to the initialization by experimenting with three natural initializations:
>
> - Unfulfill (441x Speedup): initially, all orders are unfulfilled. This setting corresponds to the theoretical results and experiments currently in the manuscript.
> - Naive (358x Speedup): Initially, fulfill all orders from their nearest fulfillment center, ignoring inventory and capacity constraints.
> - Random (390x Speedup): Initially, fulfill all orders from a random FC.
>
> Clearly, different initialization methods have an impact on the simulation speed, although massive speedups are possible regardless.
>
> Here, it appears that the "Unfulfill" strategy works the best. The intuitive explanation for this is consistent with our analysis: using this initialization strategy, after the first iteration, each processor will have correctly accounted for inventory and capacity consumed by orders for that processor's own products -- in some sense, a "greedy" initialization which is less naive than the "Naive" approach above. Further, unlike the alternatives, when initializing each chunk, this procedure is able to account for inventory and capacity consumed in previous chunks. We will add this discussion to the appendix.
>
>
> ### 2. Convergence of action sequences
> We observe similar convergence behavior for the RMSE of actions as we do for states. This should intuitively be the case as we would expect reasonable controllers to exhibit some sort of Lipschitz continuity, especially over regions of the state space for which we have training data.
>
> ### 3. Understanding convergence on MuJoCo
>
> Thank you for highlighting this interesting observation. We agree that understanding the factors affecting convergence in continuous control settings is an exciting direction.
>
> As a first step towards this kind of analysis, we can consider linear control of a linear dynamical system. Writing out the Picard iteration explicitly, with each timestep assigned to a different processor, this turns out to be equivalent to solving a particular linear system via fixed point iteration. More precisely, letting $X^{(k)} \in \mathbb{R}^{T \times d}$ be the matrix of state variables at iteration $k$, across all times $t \in [T]$, the Picard iteration computes $X^{(k+1)} = A X^{(k)} + b$ for some appropriately shaped $A, b$ which depend on the the matrices describing both the dynamics and the controller. It is then straightforward to show that $X^{(k)}$ converges geometrically at a rate depending on the spectral radius of the matrix $A$.

---

> > ### Comment · Reviewer_D3aU · 2025-04-02
> >
> > Thank you for addressing all my concerns. I've updated my review and increased my score accordingly.

---

### Decision · Program_Chairs · 2025-05-01

**Decision:**

Accept (poster)

**Comment:**

This paper introduces the Picard Iteration algorithm to accelerate policy simulation in reinforcement learning for supply chain optimization (SCO) problems, where simulating a single trajectory can take hours due to the serial nature of policy evaluations. The authors prove that for supply chain problems, this approach converges in a number of iterations independent of the time horizon, resulting in significant speedups. Empirical results demonstrate a 400x speedup on large-scale fulfillment optimization problems using a single GPU compared to sequential evaluation, and a 100x improvement over the Time Warp algorithm.

The reviewers unanimously agree that the paper is well-written, and its contributions merit acceptance. After carefully reading the rebuttal/discussion, I tend to agree. Please incorporate the following comments in the final version of the paper. In particular, addressing the following concerns will help strengthen the current version of the paper:
- Use the additional page to include some details about the implementation challenges and the effect of initialization (response to Rev. D3aU)
- Include the limitations and the worst-case setup for the Picard iteration (response to Rev. UC5Y)
- Restructure the introduction according to the suggestion of Rev. 4umB